# Farmers’ Preference for Participating in Rural Solid Waste Management: A Case Study from Shaanxi Province, China

**DOI:** 10.3390/ijerph192114440

**Published:** 2022-11-04

**Authors:** Wenyan Wu, Lu Li, Hanxin Chen, Minyue Xu, Yalin Yuan

**Affiliations:** 1College of Economics and Management, Northwest A&F University, Yangling, Xianyang 712100, China; 2School of Economics and Resource Management, Beijing Normal University, Haidian, Beijing 100875, China

**Keywords:** rural solid waste management, farmers’ preference, participation mode, social trust, institutional trust, interpersonal trust, environmental awareness

## Abstract

Rural solid waste management is an important method to improve rural living environments. Farmers’ participation in rural solid waste management plays an essential role in sustainable waste management. Based on the micro-survey data of 592 farmers in Shaanxi province, a multinomial logit model was applied to explore farmers’ preferences for participating in rural solid waste management. The empirical results show that both institutional and interpersonal trust have significant positive effects on farmers’ payment participation, and labor and payment participation. Among environmental awareness, farmers with pro-environmental behavior prefer labor participation, and labor and payment participation; the more environmental knowledge farmers have, the stronger their preferences for payment participation, labor and payment participation, and labor participation; farmers concerned about environmental problems are more inclined to adopt labor and payment participation or payment participation. For socio-demographic characteristics, gender has no significant influence, while agricultural net income and education can significantly increase farmers’ willingness to participate; farmers who have migrant work experience prefer to participate in payment; there are obvious intergenerational differences in the influence of social trust and environmental awareness on farmers’ participation preference. Therefore, providing diversified participation modes, creating a good social trust environment, and enhancing farmers’ awareness of environmental care are important in promoting rural solid waste management.

## 1. Introduction

Rural solid waste management (RSWM) is an emerging issue for solving waste pollution in developing and transitional countries worldwide, and China, as the largest developing country in the world, is no exception [1,2]. Rural solid waste pollution seriously damages the rural ecological environment in low- and middle-income countries [3]. In the past, a considerable amount of rural solid waste (especially organic waste) was recycled as food for livestock or fertilizer for agriculture, resulting in little pollution of the surrounding environment [4,5]. In recent years, rapid urbanization and the continuous improvement in rural residents’ consumption levels have contributed to the increase in rural solid waste in China [6]. It is reported that 300 million tons of solid waste are generated each year in rural China, and it continues to grow at a rate of approximately 8% to 10% per year [7]. However, affected by policy imbalances and farmers’ abilities and willingness to pay, China attaches much less importance to RSWM services in rural areas than in urban areas. According to the survey, 40% of rural towns still lack sufficient waste collection and disposal facilities and are mainly concentrated in the western regions [8,9]. Rural solid waste treatment methods in China’s vast areas generally include unregulated dumping, burning waste in open sites, and simple composting, which results in nearly 60% of the solid waste directly entering the land or water, causing severe pollution of 24% of drinking water and 18% of lake water in China [7]. Therefore, to completely solve the problem of rural solid waste, it is necessary to collect and manage rural solid waste and reduce its amount at the source.

Public participation is recognized as the main path toward sustainable waste management and plays a vital role in environmental conflict management, as it can bridge the gap between the government and citizens [9,10,11]. However, the long-term government-led governance model of the Chinese RSWM ignores the importance of farmers’ participation, resulting in high governance costs, low efficiency, and low actual participation of farmers [9,12,13]. To effectively ensure farmers’ participation in RSWM, their participation preferences and influencing factors must be clearly understood. 

Farmers’ participation in RSWM is a process in which many independently choose to participate in collective action based on the region [14]. However, the results of individual rational choice and collective rational choice are not necessarily consistent, which makes collective action a dilemma [9]. However, China’s rural areas are a society of acquaintances, and the social trust formed by farmers’ long-term interactions in villages can combine farmers’ micro-individual behaviors with macro-collective actions [15,16]. This type of trust in social relationships plays an important role in influencing farmers’ behavior [17,18]. Additionally, many studies have shown that a higher level of environmental concern significantly impacts individuals’ environmental protection willingness and environmental protection behavior [12,19]. The environmental awareness of individual farmers can be transmitted to affect their social circles through the social trust network of farmers, which can effectively promote the success of collective actions [20].

In addition, with the development of the current rural society and changes in the population structure, farmers are gradually divided into two different groups: The old generation and the new generation, which are very different in terms of life background, personal growth experience, resource endowment, social networks, interests and needs, and values [21,22]. Research shows that the willingness of the older generation of farmers to participate is lower than that of the new generation, and there are significant intergenerational differences [23]. Therefore, it is foreseeable that generational differences will inevitably lead to the diversification of farmers’ attitudes towards RSWM.

In the current literature, most research focuses on whether farmers are willing to participate in RSWM and farmers’ willingness to pay [12,24,25]. Some studies in developing countries have found that social trust, environmental awareness, knowledge, and attitudes are important factors affecting residents’ willingness to sort waste and pay [14,17,18]. Moreover, the convenience of waste collection facilities significantly affected farmers’ willingness to work [23]. Among the socioeconomic factors, gender, age, education level, income, employment status, and government policies significantly impact farmers’ willingness to participate in household waste management [9,26,27]. Few studies have focused on the multiple public preferences for participating in RSWM and its influencing factors, and the importance of these factors has not yet been well evaluated. 

Based on this, the overall goal of this research is to use Shaanxi Province as an example to describe farmers’ participation preference for RSWM and to explore the mechanism of social trust and environmental awareness on farmers’ preference for participating in RSWM. Meanwhile, with the development of the current rural society and changes in the population structure, farmers are gradually divided into the old and new generations. It is worth exploring whether there are significant differences in participation preferences between these two groups of farmers. The information garnered from this study can provide a theoretical reference for the government to promote farmers to participate in RSWM, deepen their self-management and self-service, and realize rural environmental autonomy. The remainder of this paper is organized as follows. Section 2 provides a literature review and the hypotheses. Section 3 describes sampling, data collection, and variable definitions. The results are presented in Section 4. Finally, we conclude the study in Section 5.

## 2. Literature Review

Agricultural economic issues have made great progress in sociological research since Pierre Bourdieu, James Coleman, Robert Putnam, and other scholars proposed the innovative theory of social capital [28,29]. Social trust is the core representative of social capital and is considered an important factor in rural development [28]. The higher the social trust, the more likely farmers are to participate in environmental protection cooperation. Research has also shown that social trust impacts the promotion of citizens’ environmental responsibility behavior, and it also significantly impacts public participation in tackling climate change [30,31]. A decline in the level of social trust leads to insufficient social capital stock, which will have a negative impact on environmental governance [32]. Chinese rural areas are societies of acquaintances, and social trust formed by long-term interaction is an important part of rural social capital [15,16]. Social trust effectively manages public resources from the informal system, compensates for market deficiencies, and affects farmers’ willingness to participate in environmental protection. Furthermore, social trust plays a key role in influencing pro-environmental attitudes and promoting individuals’ preferences for improving environmental quality [16,33]. A survey of Indian villages found that social trust plays a positive role in rural public decision-making and can promote villagers’ management of village public affairs [34]. The dilemma of environmental governance lies in how to overcome externalities and information asymmetry, and a high degree of social trust will help farmers overcome the expected uncertainty and information asymmetry to a certain extent because they believe that people in their social network will take measures to protect the environment as they do themselves, which is conducive to improving the efficiency of rural environmental protection governance. 

Social trust can be divided into institutional and interpersonal trust [35]. Institutional trust is a functional social mechanism embedded in special social structures, laws, and systems, and it is caused by the social phenomenon of “non-interpersonal” relationships [28]. The soft restraint mechanism formed by institutional trust based on “non-interpersonal” relationships can regulate rural social order and effectively restrain the emergence of a “free rider” and other phenomena [36,37]. There is a wealth of evidence that people are more likely to obey the law if they believe that law enforcement is fair and thus trust the government. Some scholars have shown that the higher the recognition and trust of farmers in the relevant systems and policies of rural domestic waste management, the better the prospects of garbage management, the higher the breadth and depth of participation, and the greater the possibility of cooperation [38]. Interpersonal trust is a short-radius internal relationship network based on the “emotional relationship” between people and is based on consanguinity, kinship, and geography, representing an individual’s sense of belonging and confidence in society. Previous studies have found that interpersonal trust plays a distinct role in promoting farmers’ environmental governance cooperation [39]. To a certain extent, interpersonal trust determines whether farmers are willing to trust others or rely on others’ suggestions for household waste classification. Therefore, farmers’ interpersonal trust can improve their level of domestic waste management.

Public environmental awareness is one of the most important indicators of national civilization [40]. It is the information people have about any phenomena related to their environment, their concern about the environment, and their willingness to participate in favor of the environment, including the behavior derived from that commitment [41,42]. Many environmental problems and their consequences are the result of ignorance [43]. Therefore, the public must be aware of environmental issues, their consequences, and the actions that must be taken to address these issues [44]. Environmental awareness has led to an increasing number of individuals engaging in environmentally friendly behaviors in their daily lives [45,46]. Over the last few decades, researchers in different countries have studied people’s environmental awareness. Some research results show that residents’ environmental knowledge and information are significantly related to their environmental behavior, and the lack of related knowledge and information hinders residents’ participation in waste separation and recycling [47,48]. Environmental knowledge has long been recognized as one of the most crucial factors influencing household solid waste disposal. Karkanias et al. [49] emphasized the importance of knowledge and economic incentives as a driving force for composting; when individuals have more information about the benefits of waste sorting, they are more likely to practice it. Abstract knowledge (general knowledge about waste management) is significantly correlated with willingness to engage in waste management [48].

## 3. Material and Methods

### 3.1. Data Source

Shaanxi Province is a core agricultural area in western China with a population of more than 15 million, approximately 40.57% of whom live in rural areas [50]. The Guanzhong area is in central Shaanxi Province and includes one district and five cities. Its agricultural population accounts for 80% of the total agricultural population of Shaanxi province [50]. This study selected the Guanzhong area as a representative sample to study farmers’ preferences for RSWM. Yangling District, Dali County, and Chengcheng County of Weinan City, and Taibai County of Baoji City (Figure 1) were selected for the survey based on a stratified and random sampling method of “district (county)–township–natural village–farming household.” Each district (county) selected 1–2 towns/streets, each town/street selected 4–5 natural villages, and each natural village randomly sampled 30–40 farmers for face-to-face interview surveys. Finally, we approached 600 households and completed 592 interviews, with a 98.67% response rate.

### 3.2. Sample Characteristics

Table 1 shows respondents’ socioeconomic characteristics. The interviewed farmers were mostly over 50 years old (55.1%), related to the migration of young rural laborers to cities and the return of old and weak people to rural areas. The proportion of male and female respondents in the sample was balanced, with 51% male and 49% female. Regarding education, most respondents had a low education level, with only 21.8% having higher education (12 years of education or more). Of the respondents, 51.5% had experience going out to work. This is more in line with the current reality of rural China’s outflow of young and middle-aged labor. The disposable personal income of respondents in 2018 was less than 10,000 yuan (69.7%), which was generally low. The survey results are similar to the per capita disposable income of rural residents of 11,213 yuan in the “Shaanxi Provincial Statistical Yearbook 2019” [50], indicating that this sample has a certain representativeness. The details are presented in Table 1.

### 3.3. Measurement

#### 3.3.1. Dependent Variables

The dependent variable in the analysis is farmers’ preference for participating in the RSWM. Some studies show that more than two-thirds of the respondents were willing to pay for RSWM services in the rural areas of West China, which can alleviate the financial pressure on the village [25]. Some interviewees stated they were willing to deliver their domestic waste to the collection facilities installed in local villages [23]. This method of labor participation can improve the efficiency of garbage collection. In addition, some studies analyzing why farmers are reluctant to participate in waste sorting have found that excessive fees or labor intensity can reduce farmers’ enthusiasm for waste sorting, suggesting that an appropriate combination of labor and costs is established [9,23].

According to these studies, this study sets the question of measuring farmers’ preference for participating in RSWM as “RSWM can improve ecological and environmental service functions. If the government wants to improve the environment, solid waste will be uniformly recycled and disposed of. As an environmental beneficiary, please choose according to your preference.” The answer options were *labor participation* (LP), *payment participation* (PP), *labor and payment participation* (LPP), and *nonparticipation* (NP).

LP: Farmers can choose to transport community waste to collection points, participate in cleaning up collection points to keep the dump clean, or supervise others to ensure they clean up as required.

PP: Cleaners are responsible for the transportation, cleaning, and supervision of waste management, and farmers pay for the employment of cleaners.

LPP: Farmers are willing to participate in labor and pay for hiring cleaners for waste management.

NP: Farmers are not willing to participate in labor or pay for waste management.

Figure 2 shows the distribution of preference options for waste management. The highest percentage of respondents (62%) indicated they preferred to participate in waste management through the LPP. This is encouraging because these respondents were willing to participate in waste management through a mixed approach. However, another 21% indicated that their intention was only to pay for waste management because it was too troublesome to participate in labor.

#### 3.3.2. Independent Variables

Referring to the method of sociologists, social trust was divided into institutional and interpersonal trust [35]. This study used interviewees’ trust in relatives, neighbors, and cleaners to conduct factor analysis to obtain interpersonal trust. Concerning institutional trust, respondents were asked to express their trust degree with the following statements: “I support rural household waste management policy”; “I think the current waste management policy is reasonable”; and “I trust the village cadres.” All questions in this questionnaire were measured using a 5-point Likert scale ranging from strongly disagree to strongly agree.

The Kaiser–Meyer–Olkin (KMO) test was used to measure sampling adequacy and Bartlett’s test of sphericity was applied to the factor analysis applicability test. This study used SPSS 19.0 software (IBM SPSS Statistics, USA) to conduct factor analysis on institutional trust and interpersonal trust, respectively. The results show that the KMO values of institutional trust and interpersonal trust are 0.844 and 0.750, respectively, which are greater than 0.6, and the p value for the statistical significance of Bartlett’s test of sphericity is 0.000. p < 0.001, indicating that the factor analysis data had good validity and were suitable for factor analysis. 

Environmental awareness refers to the information people possess about any phenomena related to their environment, their concerns about the environment, and their willingness to act in favor of the environment. This study used environmental behavior, environmental knowledge, and environmental attitudes to measure environmental awareness. The questions with a 5-point Likert scale ranging from strongly disagree to strongly agree in the questionnaire were “I often sort the garbage and deliver it correctly,” “I often collect waste separation knowledge and information”, and “I am very concerned about environmental problems”.

Therefore, following hypotheses are proposed:

**Hypothesis 1.** 
*Institutional trust can significantly improve farmers’ preference for participating in RSWM.*


**Hypothesis 2.** 
*Interpersonal trust can significantly improve farmers’ preference for participating in RSWM.*


**Hypothesis 3.** 
*Environmental behavior can significantly improve farmers’ preference for participating in RSWM.*


**Hypothesis 4.** 
*Environmental knowledge can significantly improve farmers’ preference for participating in RSWM.*


**Hypothesis 5.** 
*Environmental attitude can significantly improve farmers’ preference for participating in RSWM.*


It is widely accepted that socioeconomic and demographic factors affect farmers’ environmental willingness and pro-environmental behavior [19,27,48,51]. Therefore, this study used gender, age, education, net agricultural income, and migrant work experience as control variables. The definitions and descriptive statistics of these variables are presented in Table 2.

### 3.4. Multinomial Logit Models

The dependent variables in this study consist of four categories: LP, PP, LPP, and NP. To explain this set of polytomous answers, we relied on random utility theory: When faced with J alternatives, respondent i prefers alternative k if the utility they derive from it is at least as large as the utility they derive from any other alternative:(1)$$Pr_{i}\left(k\right)=Pr\left(U_{ik}\geq U_{ij},\forall j\in \left\{1,\;\ldots ,J\right\}\right)$$

If $U_{ij}$ is partitioned into a component explainable by the modeler (denoted by $V_{ij}$) and a component that is not (denoted by $\varepsilon_{ij}$), then Equation (1) [52] is equivalent to that of:(2)$$Pr_{i}\left(k\right)=\mathrm{Pr}\left(U_{ik}\geq U_{ij},\;\forall j\in \left\{1,\;\ldots ,J\right\}\right)=Pr\left(\varepsilon_{ij}-\varepsilon_{ik}\leq V_{ik}-V_{ij},\;\forall j\in \left\{1,\;\ldots ,J\right\}\right)$$

Assuming that errors are independently and identically Weibull distributed, we obtain the multinomial logit model (MNL) [52]. For that model, the odds [53] that respondent i will choose alternative k over alternative j can be expressed as:(3)$$Pr_{i}\left(k\right)/Pr_{i}\left(j\right)=\exp\left(V_{ik}\right)/\exp\left(V_{ij}\right)=\exp\left(X_{i}\beta_{k|j}\right)$$
where the last equality holds if this further assumes that:(4)$$V_{ij}=X_{i}\beta_{j}^{\prime }$$
where $X_{i}$ is a vector of variables characterizing respondent i and $\beta_{j}$ is a vector of unknown coefficients pertaining to alternative j, which can be estimated by maximum likelihood. $X_{i}$ included social trust, environmental awareness, and socioeconomic variables.

Equation (3) is the independence of irrelevant alternatives (IIA), which means that the probability ratio of the choice does not depend on the characteristics of other choices. This must be verified as a good illustration of the MNL. To test the IIA assumption, two common tests, Hausman [54] and Small-Hsiao [55], were adopted in this study. In addition, common statistical tests were performed to assess the validity of the modeling assumptions.

## 4. Results and Discussion

Table 3 displays the MNL results for farmers’ preferences in RSWM management. The MNL models were estimated using Stata 16.0 software (StataCorp LLC, College Station, TX, USA). This study used post-estimation analysis and examined measures of fit; as a result, this model is correctly specified. Then, we checked independent variables for multicollinearity and examined a variety of interaction terms [53,56]. In the end, the Hausmane–McFadden test and the Smalle–Hsiao test were used to test the assumptions of the independence of irrelevant alternatives (IIA), and these tests failed to reject the null hypothesis that the IIA holds. 

The results in Model 1 in Table 3 indicate that social trust has a positive effect on farmers’ preferences for participating in RSWM. In social trust, institutional and interpersonal trust have a significant positive effect on farmers’ LPP and PP. Under the same conditions, people who trust institutions believe that their announced environmental policies will be implemented in practice, increase their willingness to participate in RSWM, and are more willing to contribute. It follows that the more farmers recognize and trust RSWM policies, the more they will manage waste separation in the future, the more they will be aware of separation, and the more willing they will be to participate in RSWM [57]. Simultaneously, farmers prefer LPP, followed by PP, which indicates that the higher the interpersonal trust, the stronger farmers’ preference for participatory environmental management, mainly because rural China is a typical “acquaintance society,” which helps reduce the adverse effects of information asymmetry, gives full play to the connecting effect of interpersonal trust, and promotes farmers’ cooperation in environmental protection [9].

We obtained a positive and statistically significant coefficient on the environmental behavior variable, which was significantly associated with LP and LPP, suggesting that farmers who have a habit of sorting and delivering garbage are more likely to participate in the labor process. Environmental knowledge positively contributes to PP and LPP, followed by LP. The more environmental knowledge and information farmers have, the stronger their PP and LPP are, followed by LP. Environmental attitudes have a significant positive effect on LPP and PP. This further suggests that farmers’ environmental awareness is significantly associated with their preference to participate in RSWM and that a lack of relevant knowledge and information will reduce residents’ preference to participate in waste separation and recycling [58]. 

Regarding the socioeconomic and demographic characteristics considered, gender has no significant influence on farmers’ preferences for participating in RSWM. This result may be due to the state’s promotion of domestic waste management methods in rural areas, which resulted in no significant difference in awareness between male and female residents [9]. It was worth noting that there were statistically significant negative effects between age and farmers’ willingness to participate. The older the farmers, the less willing they are to participate in RSWM. A possible reason for this is that as farmers age, their ability to work and financial income decrease, so they are reluctant to participate in RSWM [23]. This study divides farmers into two groups based on age, the new generation of farmers and the older generation of farmers [21,22,23], and regresses the groups to test this conjecture. Meanwhile, families with a higher net agricultural income have stronger LPP, followed by PP and LP. This might be because rural families with a high household agricultural net income rely on agricultural production for their income and are highly dependent on rural areas. As a result, they have a stronger desire to participate in solving surrounding environmental problems. EDU has a significantly positive impact on farmers’ partition willingness; moreover, LP is in first place, LPP is close behind, and PP is in last place, which indicates that education can enhance social responsibility and improve participation. This is consistent with the finding that higher-educated individuals are more likely to participate in environmental protection [24]. MWE had a statistically significant positive impact on PP. Compared to respondents with no migrant work experience, farmers with migrant work experience are more likely to participate in domestic waste management by PP. A possible reason is that due to the early initiation of urban waste management, farmers who have city work experience are exposed to the environmental benefits of domestic waste management earlier than those who do not, which affects farmers’ willingness to participate [25]. However, due to the high opportunity cost of labor, they are more inclined to choose to pay for participation. 

To test the robustness of the MNL model, we used a multivariate probit (MVP) model (Model 2 in Table 3) for further verification, and the significance of the key variables did not change significantly. This indicates that the data analysis results of Model 1 are robust.

At present, when the aging of the agricultural population is a widespread concern, there are material or emotional differences in the relationship between the old and new generations of farmers and the countryside due to differences in their historical backgrounds, living environments, and educational levels during their upbringing. Therefore, it can be predicted that intergenerational differences will inevitably lead to the diversification of farmers’ attitudes towards household waste management. The 1980s are usually taken as the dividing line between new and old generations of farmers. We consider the lag in the formation of values in the “generation effect.” On this basis, this study extends for five years, taking birth in 1975 as the boundary, thus forming virtual variables that reflect intergenerational differences. If the interviewees were born before 1975, they were the older-generation farmer group; otherwise, they were the new-generation farmer group [21,22,23]. 

Table 4 shows the intergenerational differences in farmers’ willingness to participate in RSWM, indicating significant differences in the willingness to participate between the new and older generations of farmers. In the new-generation farmer group, institutional trust ($S_{1}$) had a significant positive impact on farmers’ PP. The higher the trust of the new generation of farmers in RSWM institutions, the stronger their PP, followed by LPP. Environmental behavior ($E_{1}$) has a significant positive impact on new-generation farmers’ PP. The new generation of farmers with pro-environmental behaviors is more willing to pay to participate in PP. One possible reason for this is that the payment method is more convenient [9]. Environmental knowledge ($E_{2}$) had a positive and significant impact on farmers’ PP and LPP. The more knowledge and information the new generation of farmers have about waste classification, the stronger their PP, followed by LPP.

In the older-generation farmer group, interpersonal trust ($S_{2}$) affected farmers’ LP and LPP. The older generation of farmers with high interpersonal trust had higher LP, followed by LPP. Environmental behavior ($E_{1}$) is closely related to the willingness of older farmers to participate. The older generation of farmers with pro-environmental behaviors is more willing to work (LP), followed by LPP. Overall, the older generation of farmers is more willing to participate in RSWM through labor. This may be because the older generation of farmers has accumulated a high degree of interpersonal trust in their long-term rural life and are generally aware of conservation [59,60]. Meanwhile, most of the older generation of farmers have stopped working or shortened their working hours, so they have plenty of time, but they are not financially strong [61]. Therefore, LP was stronger, followed by LPP. 

## 5. Conclusions

Broad and sustainable public participation forms the basis for successful rural environmental management. Satisfaction with participation preferences is a key factor in participation. The present study proposed that farmers’ preference for participating in RSWM showed that the highest proportion of households (62%) prefer contributing both labor and payment for RSWM, followed by payment only (21%). Both dimensions of social trust have significant positive effects on farmers’ preference for LPP and PP, but not LP. In environmental awareness, pro-environmental behavior improves farmers’ preference for LP and LPP; environmental knowledge enhances farmers’ willingness to participate; and a positive environmental attitude will make farmers more inclined to adopt the LPP or PP approach. Furthermore, there were obvious intergenerational differences in the influence of social trust and environmental awareness on farmers’ willingness to participate.

## 6. Implications

The following political implications can be drawn from this study: First, the government should set up diversified participation and incentive approaches to meet the needs of farmers with different characteristics. Second, in the rural revitalization strategy, the grassroots government should enhance farmers’ sense of belonging and community cohesion through various forms (such as continuously holding community activities and attracting social attention with the help of self-media platforms), give full play to the implicit incentive function of social trust, increase farmers’ willingness to participate, and promote the effective implementation of waste management policies. Last but not least, governments at all levels, rural communities, and commonwealth organizations should continue to give full play to the main role of environmental education and propaganda and strive to build an eco-friendly socio-cultural context to create favorable conditions for the public to deeply understand, accept, and participate in rural solid waste management.

Although this study revealed the preferred mode of farmers participating in RSWM and the mechanism of influencing factors, it should be noted that the length of labor participation, the level of payment, and the relationship between the two are not identified. Moreover, the comparatively small sample size and selected location might limit the generalization of our findings. Therefore, future research should be further explored in wider communities.

## Figures and Tables

**Figure 1 ijerph-19-14440-f001:**
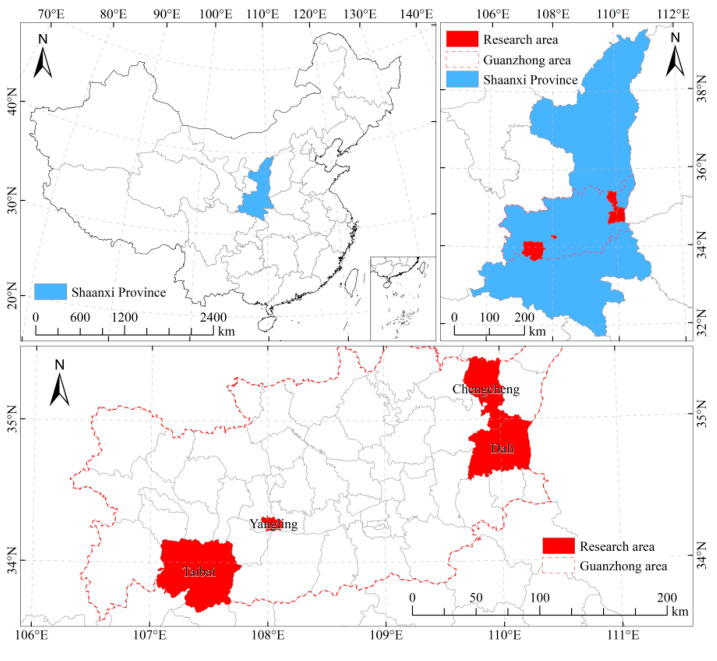
Study area.

**Figure 2 ijerph-19-14440-f002:**
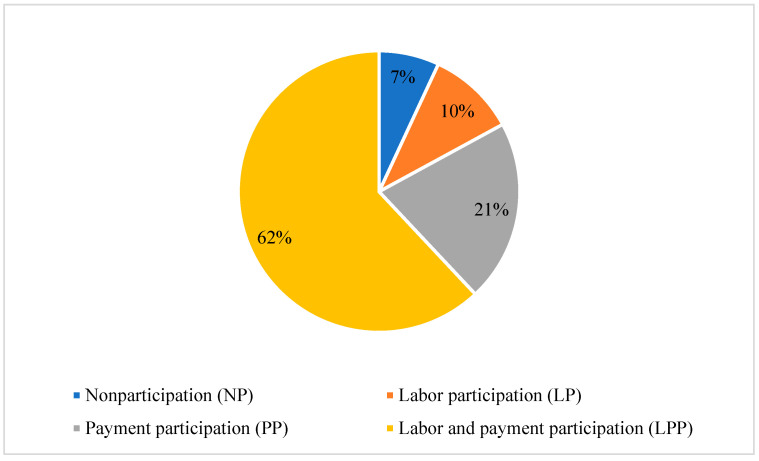
Distribution of participation preferences for rural solid waste management.

**Table 1 ijerph-19-14440-t001:** Socioeconomic characteristics of the respondents.

Item	Level	Freq.	%	Item	Level	Freq.	%
Age	A ≤ 30	91	15.4	Migrant work experience	Yes	305	51.5
	30 ˂A ≤ 50	175	29.6		No	287	48.5
	50 ˂ A ≤ 70	284	48.0	Permanent residents (in person)	*p* ≤ 2	200	33.8
	A ˃ 70	42	7.1		3 ≤ *p* ≤ 4	201	34.0
Gender	Female	306	51.7		*p* ≥ 5	191	32.3
	Male	286	48.3	Disposable personal income (thousand yuan)	D ≤ 5	224	37.8
Education (in years)	E ≤ 5	112	18.9		5 ˂ D ≤ 10	189	31.9
	6 ≤ E ≤ 8	143	24.2		10 ˂ D ≤ 15	76	12.8
	9 ≤ E ≤ 11	208	35.1		15 ˂ D ≤ 20	41	6.9
	E ≥ 12	129	21.8		D ˃ 20	62	10.5

**Table 2 ijerph-19-14440-t002:** Variables included in the multinomial logit model and coding of levels.

Variables		Description	Mean	SD
Dependent variable				
Participation preference		Nonparticipation = 0, labor participation = 1, payment participation = 2, labor and payment participation = 3	2.380	0.925
Core Independent variables				
Social trust	Institutional trust ($S_{1}$)	Calculated based on factor analysis	3.890	0.891
	Interpersonal trust ($S_{2}$)	Calculated based on factor analysis	4.650	0.682
Environmental awareness	Environmental behavior ($E_{1}$)	I often sort the garbage and deliver it correctly: N	2.441	0.806
	Environmental knowledge ($E_{2}$)	I often collect waste separation knowledge and information: N	1.990	1.025
	Environmental attitude ($E_{3}$)	I am very concerned about environmental problems: N	4.240	0.874
Control variables				
Gender (GEN)		Male = 0, female = 1	0.480	0.500
Age (AGE) (in years)		0 ˂ AGE ≤ 30 = 1, 30 ˂ AGE ≤ 60 = 2, AGE ˃ 60 = 3	2.400	0.740
Agricultural net income (ANI) (Ten thousand yuan)		Household net income from agriculture in 2018: ANI = 0 = 0; 0˂ ANI ≤ 1 = 1; 1˂ ANI ≤ 2 = 2; 2˂ ANI ≤ 3 = 3; ANI ˃ 3 = 4	0.976	1.110
Education (EDU)		EDU ≤ 5 = 1, 5 ˂ EDU ≤ 8 = 2, 9 ≤ EDU ≤ 11 = 3, EDU ≥ 12 = 4	2.598	1.028
Migrant work experience (MWE)		I have migrant work experience: no = 0, yes = 1	0.520	0.500

Note: “N” means “strongly disagree = 1, disagree = 2, generally = 3, agree = 4, and strongly agree = 5”.

**Table 3 ijerph-19-14440-t003:** Results of multinomial logit model for farmers’ preference for participating in rural solid waste management.

Variable	MNL (Model 1)			MVP (Model 2)		
	LP	PP	LPP	LP	PP	LPP
$S_{1}$	0.311	0.410 *	0.564 ***	0.217	0.272 *	0.399 ***
	(0.241)	(0.220)	(0.207)	(0.154)	(0.143)	(0.137)
$S_{2}$	0.184	0.383 **	0.668 ***	0.142	0.266 **	0.495 ***
	(0.185)	(0.176)	(0.167)	(0.126)	(0.121)	(0.117)
$E_{1}$	0.425 *	0.334	0.424 **	0.246 *	0.192	0.271 *
	(0.262)	(0.235)	(0.221)	(0.168)	(0.155)	(0.149)
$E_{2}$	0.512 *	0.669 **	0.668 **	0.376 **	0.488 ***	0.498 ***
	(0.291)	(0.271)	(0.263)	(0.183)	(0.173)	(0.169)
$E_{3}$	0.046	0.335 *	0.381 **	0.018	0.202 *	0.250 *
	(0.222)	(0.210)	(0.194)	(0.145)	(0.142)	(0.133)
GEN	0.239	0.266	0.532	0.255	0.247	0.467
	(0.443)	(0.406)	(0.384)	(0.285)	(0.266)	(0.256)
AGE	−0.959 **	−0.819 **	−1.070 ***	−0.54 **	−0.442 **	−0.645 ***
	(0.392)	(0.373)	(0.359)	(0.235)	(0.224)	(0.218)
ANI	0.861 ***	0.815 ***	0.905 ***	0.481 **	0.450 **	0.533 ***
	(0.332)	(0.309)	(0.297)	(0.202)	(0.189)	(0.183)
EDU	0.992 **	0.861 **	0.888 **	0.627 **	0.543 **	0.573 **
	(0.453)	(0.417)	(0.396)	(0.287)	(0.269)	(0.259)
MWE	0.096	0.671 *	0.552	0.037	0.419 *	0.353
	(0.450)	(0.407)	(0.386)	(0.282)	(0.262)	(0.251)
Log likelihood	−560.001			−559.889		
$\mathrm{LR}\;\mathrm{value}(p\;>\;chi^{2}$)	112.270 (0.000)			—		
$\mathrm{Pseudo}\;R^{2}$	0.091			—		
$\mathrm{Wald}\;\mathrm{test}(p>\;chi^{2}$)	—			85.881 (0.000)		
N	592			592		

Notes: The base outcome is “nonparticipation”; *, **, *** refer to *p* ≤ 0.10, *p* ≤ 0.05, and *p* ≤ 0.01, respectively; the value in parentheses is a standard error.

**Table 4 ijerph-19-14440-t004:** Analysis of intergenerational differences.

Variable	New Generation (Model 3)			Older Generation (Model 4)		
	LP	PP	LPP	LP	PP	LPP
$S_{1}$	2.790	2.547 **	2.489	0.463	0.503	0.695
	(1.488)	(1.478)	(1.470)	(0.271)	(0.243)	(0.225)
$S_{2}$	2.098	2.581	2.554	0.087 **	0.253	0.601 **
	(1.178)	(1.185)	(1.171)	(0.211)	(0.195)	(0.184)
$E_{1}$	2.977	3.500 ***	2.814	0.585 **	0.695	0.496 *
	(2.192)	(2.182)	(2.179)	(0.295)	(0.264)	(0.240)
$E_{2}$	2.672	2.884 ***	2.937	0.464	0.628	0.587
	(1.405)	(1.394)	(1.389)	(0.326)	(0.296)	(0.281)
$E_{3}$	0.467	1.126	0.964	0.183	0.272	0.445
	(0.828)	(0.833)	(0.818)	(0.267)	(0.239)	(0.220)

Note: *** significant at 1%, ** significant at 5%, * significant at 10%; the value in parentheses is a standard error; the reference category is nonparticipation; limited to space, the table shows only the regression results of key variables.

## Data Availability

The dataset used in the path analyses and the full questionnaire are available from the corresponding author upon request.

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
