# Peer review of "Farmers’ Preference for Participating in Rural Solid Waste Management: A Case Study from Shaanxi Province, China"

_ijerph, 2022, doi:10.3390/ijerph192114440_

Round 1

Reviewer 1 Report

This work reports an interesting and well presented study about farmers’ preferences for participating in rural solid waste management based on a micro-survey data of 592 farmers in Shaanxi province (China). I suggest it for publication after these major revision:

1. in the "Conclusions and Implication" paragraph, there are no references to any limitations of the study. I suggest you include a few sentences about it.

2. in the "Conclusions and Implication" paragraph, there are no reference to comparison with the results of other similar studiesI suggest reviewing this paragraph paying more attention to such comparisons

Minor revision

1. sentences in lines between 13 and 20 are tiring to read and understand. I suggest rearranging them for easier reading

Author Response

Response to Reviewer 1 Comments

We are grateful to Reviewer 1 for your critical evaluation of our manuscript entitled “Farmers’ Preference for Participating in Rural Solid Waste Management: A case study from Shaanxi Province, China”. Your comments were very valuable and helpful to us. We have carefully revised our manuscript according to your comments and suggestions. We sincerely hope that the revised version of our manuscript meets the requirements for publication in IJERPH. If there are any further comments and suggestions, please let us know. We will be very pleased to modify our manuscript accordingly.

Point 1: In the "Conclusions and Implication" paragraph, there are no references to any limitations of the study. I suggest you include a few sentences about it.

Response 1: Thank you very much for your valuable suggestion. In the "Conclusions and Implication" paragraph, we have added this study’s limitation, which is “Although this study revealed the preferred mode of farmers participating in RSWM and the mechanism of influencing factors, it should be noted that the length of labor participation, the level of payment, and the relationship between the two are not identified. Moreover, the comparatively small sample size and selected location might limit the generalization of our findings. Therefore, future research should be further explored in the wider communities.” Please see page 11, line 421-426, highlighted in yellow background.

Point 2: In the "Conclusions and Implication" paragraph, there are no reference to comparison with the results of other similar studies. I suggest reviewing this paragraph paying more attention to such comparisons.

Response 2: Thank you very much for your suggestion. We have added the comparison with the results of other similar studies, such as, “gender has no significant influence on farmers’ preferences for participating in RSWM. This result may be due to the state’s promotion of domestic waste management methods in rural areas, resulted in no significant difference in awareness between the male and female residents [9]. It was worth noting that” (Please see page 9, line 328-332, highlighted in yellow background.), ” [21-23]” (Please see page 9, line 337, highlighted in yellow background.),” A possible reason is that due to the early initiation of urban waste management, farmers who have city work experience are exposed to the environmental benefits of domestic waste management earlier than those who do not have, which affects farmers’ willingness to participate [25]. (Please see page 10, line 349-352, highlighted in yellow background.)”, “the highest proportion of households (62.0%) prefer contributing both labor and payment for RSWM, followed by payment only (20.9%), which is in accordance with Yuan et al. [9]. In social trust, both institutional and interpersonal trust have significant positive effects on farmers’ preference for LPP and PP, but not LP [9, 57]. Moreover, for environmental awareness, pro-environmental behavior improves farmer preference for LP and LPP; environmental knowledge enhances farmers' willingness to participate [58];” (Please see page 10, line 398-403, highlighted in yellow background.)” Thanks again for your very good suggestion.

Point 3: Sentences in lines between 13 and 20 are tiring to read and understand. I suggest rearranging them for easier reading

Response 3: Thank you very much for your reminding. We have rewritten the sentences “The empirical results show that: both institutional trust and interpersonal trust have significant positive effects on farmers’ payment participation, and labor and payment participation; among environmental awareness, environmental behavior is significantly associated with labor participation, and labor and payment participation, and environmental knowledge positively contributes to payment participation, and labor and payment participation, followed by labor participation; environmental attitude has a significant positive effect on labor and payment participation and payment participation;” as “The empirical results show that: both institutional and interpersonal trust have significant positive effects on farmers’ payment participation, and labor and payment participation. Among environmental awareness, farmers with pro-environmental behavior prefer labor participation, and labor and payment participation; the more farmers have environmental knowledge, the stronger their preferences of payment participation, labor and payment participation, and labor participation; farmers concerned about environmental problems are more inclined to adopt a labor and payment participation or payment participation. For socio-demographic characteristics, gender has no significant influence, while agricultural net income and education can significantly increase farmers' willingness to participate; farmers who have migrant work experience prefer to participate in payment;” We hope the content now could be easier to understand. Please see page 1, line 15-24, highlighted in yellow background.

Reviewer 2 Report

Municipal solid waste especial management is a serious issue. The manuscript surveyed the Farmers’ Preference for Participating in Rural Solid Waste. The research results are meaningful for rural solid waste management. In review of the manuscript, I suggest a major revision before its publication.

Special comments

1.     The data source and data characteristics should be in the section of material and methods instead of the section of results. Meanwhile, the hypothesis should be in section of material and methods.

2.     The survey areas needs to be drawn in the map

3.     The Multinomial logit models should also be in the section of material and methods

Author Response

Response to Reviewer 2 Comments

We are grateful to Reviewer 2 for your critical evaluation of our manuscript entitled “Farmers’ Preference for Participating in Rural Solid Waste Management: A case study from Shaanxi Province, China”. Your comments were very valuable and helpful to us. We have carefully revised our manuscript according to your comments and suggestions. We sincerely hope that the revised version of our manuscript meets the requirements for publication in IJERPH. If there are any further comments and suggestions, please let us know. We will be very pleased to modify our manuscript accordingly.

Point 1: The data source and data characteristics should be in the section of material and methods instead of the section of results. Meanwhile, the hypothesis should be in section of material and methods.

Response 1: Thank you very much for your helpful comments. We have changed the section “3. Results” to “3. Material and Methods”. And the data source and data characteristics are now belonged to section 3.1 and 3.2, separately. Please see page 4, line 174-175; page 5, line 190, which are highlighted in yellow background. Hypothesis are also in the section of “3. Material and Methods”, please see page 6-7, line 250-260, which are highlighted in yellow background.

Point 2: The survey areas needs to be drawn in the map.

Response 2: Thank you very much for your suggestion. We have inserted the map of our study area (Figure 1). Please see page 4, line 176-177, which are highlighted in yellow background.

Point 3: The Multinomial logit models should also be in the section of material and methods.

Response 3: Thank you very much for your reminding and helpful suggestion. The Multinomial logit models (3.4. Multinomial logit models) have been in the section of “3. material and methods”, please see page 7, line 268, highlighted in yellow background.

Reviewer 3 Report

An author has made a good attempt to understand the "  Farmers’ Preference for Participating in Rural Solid Waste 2 Management: A case study from Shaanxi Province, China". However, few comments are mentioned below:

1. Objective has not been defined effectively

2. Existing literature is missing

3. Discussion of tables is not discussed effectively.

4. Conclusion should be rewritten effectively. 

Author Response

Response to Reviewer 3 Comments

We are grateful to Reviewer 3 for your critical evaluation of our manuscript entitled “Farmers’ Preference for Participating in Rural Solid Waste Management: A case study from Shaanxi Province, China”. Your comments were very valuable and helpful to us. We have carefully revised our manuscript according to your comments and suggestions. We sincerely hope that the revised version of our manuscript meets the requirements for publication in IJERPH. If there are any further comments and suggestions, please let us know. We will be very pleased to modify our manuscript accordingly.

Point 1: Objective has not been defined effectively.

Response 1: Thank you very much for your valuable comment. The objective has been rewritten from “Based on this, the overall goal of this research is to use Shaanxi Province as an example to describe the current status of RSWM in rural areas in western China and to explore the influence of social trust and environmental awareness on farmers’ preference for participating in RSWM to provide a theoretical reference for deepening Chinese farmers’ self-management and self-service and realizing rural environmental autonomy.” to “Based on this, the overall goal of this research is to use Shaanxi Province as an example to describe farmers’ participation preference of RSWM and to explore the mechanism of social trust and environmental awareness on farmers’ preference for participating in RSWM. Meanwhile, with the development of the current rural society and changes in the population structure, farmers are gradually divided into the old and new generations. It’s worth exploring whether there are significant differences in participation preferences between these two groups of farmers. The information garnered from this study can provide a theoretical reference for the government to promote farmer to participate in RSWM, deepen their self-management and self-service, and realize rural environmental autonomy.” Please see page 2, lines 89-98, highlighted in yellow background.

Point 2: Existing literature is missing.

Response 2: Thank you very much for your reminding. We have added “[21-23]” in page 8, line 336, highlighted in yellow background; “[25]” in page 9, line 351, highlighted in gray background; “[21-23]” in page 10, line 367, highlighted in yellow background; [9], [9,57], and [58] in page 10 lines 400-403, highlighted in gray background.

Point 3: Discussion of tables is not discussed effectively.

Response 3: Thank you very much for your reminding and valuable suggestion. We added “gender has no significant influence on farmers’ preferences for participating in RSWM. This result may be due to the state's promotion of domestic waste management methods in rural areas, resulted in no significant difference in awareness between the male and female residents [9]. It was worth noting that” (Please see page 9, line 327-331, highlighted in yellow background.), “[21-23]” (Please see page 9, line 336, highlighted in yellow background.), “which indicates that education can enhance social responsibility and improve participation” (Please see page 9, line 343-344, highlighted in yellow background.), “A possible reason is that due to the early initiation of urban waste management, farmers who have city work experience are exposed to the environmental benefits of domestic waste management earlier than those who do not have, which affects farmers’ willingness to participate [25]. However, due to the high opportunity cost of labor, they are more inclined to choose to pay for participation.” (Please see page 9, line 348-352, highlighted in yellow background.), and “[21-23]” (Please see page 10, line 367, highlighted in yellow background.) for table 3.

Point 4: Conclusion should be rewritten effectively.

Response 4: Thank you very much for your valuable comment. We have rewritten the conclusion and implications as “the highest proportion of households (62.0%) prefer contributing both labor and payment for RSWM, followed by payment only (20.9%). In social trust, both institutional and interpersonal trust have significant positive effects on farmers’ preference for LPP and PP, but not LP. Moreover, for environmental awareness, pro-environmental behavior improves farmer preference for LP and LPP; environmental knowledge enhances farmers' willingness to participate; and a positive environmental attitude will make farmers more inclined to adopt the LPP or PP approach. Furthermore, there were obvious intergenerational differences in the influence of social trust and environmental awareness on farmers’ willingness to participate.

The following political implications can be drawn from this study: First, in the rural revitalization strategy, the grassroots government should enhance farmers' sense of belonging and community cohesion through various forms (such as holding continuous community activities, attracting social attention with the help of self-media platforms), give full play to the implicit incentive function of trust, improve farmers' willingness to participate, and promote the effective implementation of waste management policies. Second, governments at all levels, rural communities, and the commonweal organizations should help farmers to understand the problem of environmental pollution in a number of ways, such as through public service advertisements, cultural publicity, education, and training, especially the impact of domestic waste on farmers’ physical and mental health, rural living environment, and the ecological environment, enhance the farmers' environ-mental awareness, cultivate the consciousness of their protagonists to participate in solid waste management. Last but not the least, the government should set up diversified participation and incentive ways to meet the needs of farmers.

Although this study revealed the preferred mode of farmers participating in RSWM and the mechanism of influencing factors, it should be noted that the length of labor participation, the level of payment, and the relationship between the two are not identified. Moreover, the comparatively small sample size and selected location might limit the generalization of our findings. Therefore, future research should be further explored in the wider communities.” Please see page 10-11, line 398-426, highlighted in yellow background.
